# IMPROVING LONG-HORIZON IMITATION THROUGH LANGUAGE PREDICTION

## ABSTRACT

Complex, long-horizon planning and its combinatorial nature pose steep challenges for learning-based agents. Difficulties in such settings are exacerbated in low data regimes where over-fitting stifles generalization and compounding errors hurt accuracy. In this work, we explore the use of an often unused source of auxiliary supervision: language. Inspired by recent advances in transformer-based models, we train agents with an instruction prediction loss that encourages learning temporally extended representations that operate at a high level of abstraction. Concretely, we demonstrate that instruction modeling significantly improves performance in planning environments when training with a limited number of demonstrations on the BabyAI and Crafter benchmarks. In further analysis we find that instruction modeling is most important for tasks that require complex reasoning, while understandably offering smaller gains in environments that require simple plans. Our benchmarks and code will be publicly released [1].

## 1 INTRODUCTION

Intelligent agents ought to be able to complete complex, long horizon tasks and generalize to new scenarios. Unfortunately, policies learned by modern deep-learning techniques often struggle to acquire either of these abilities. This is particularly true in planning regimes where multiple, complex, steps must be completed correctly in sequence to complete a task. Realistic constraints, such as partial observability, the underspecification of goals, or the sparse reward nature of many planning problems make learning even harder. Reinforcement learning approaches often struggle to effectively learn policies and require billions of environment interactions to produce effective solutions (Wijmans et al., 2019; Parisotto et al., 2020). Imitation learning is an alternative approach based on learning from expert data, but can still require millions of demonstrations to learn effective planners (Chevalier-Boisvert et al., 2019). Such high data constraints make learning difficult and expensive.

Unfortunately the aforementioned issues with behavior learning are only exacerbated in the low data regime. First, with limited training data agents are less likely to act perfectly at each environment step, leading to small errors that compound overtime in the offline setting. Ultimately, this leads to sub-par performance over long horizons that can usually only be improved by carefully collecting additional expert data (Ross et al., 2011). Second, deep-learning based policies are more likely to overfit small training datasets, making them unable to generalize to new test-time scenarios. On the other hand, humans have the remarkable ability to interpolate previous knowledge and solve unseen long-horizon tasks. After observing an environment, we might deduce plan or sequence of the steps to follow to complete our objective. However, imitation learning agents are not required to construct plans by default – they are trained to only output the direct next action given seen observations. This begs the question: how can we make agents reason better in long-horizon tasks?

An attractive solution lies in language instructions, the same medium humans use for mental planning (Gleitman & Papafragou, 2005). Several prior works directly provide agents with language instructions to follow (Anderson et al., 2018; Shridhar et al., 2020; Chen et al., 2019). Unfortunately, such approaches require the specification of exhaustive instructions at test time for systems to function. A truly intelligent agent ought to be able to devise its own plan and execute it, with only a handful of demonstrations. We propose improving policy learning in the low-data regime

---

[1]Code is available at: `<github will be made public after reviewing period>`

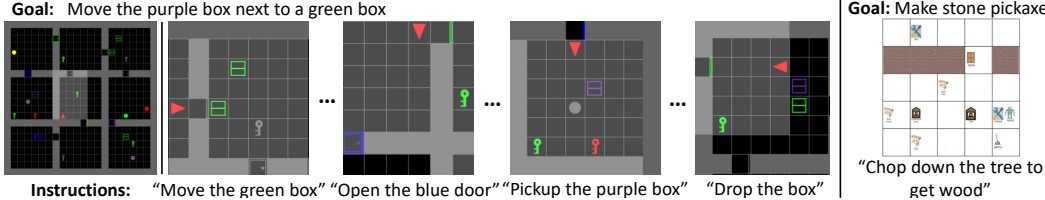

Figure 1: The left half of the figure shows key frames and their associated instructions for a task in the BabyAI environment. The right half depicts a single state from the Crafting environment and its associated goal and instruction.

by having agents predict high-level planning instructions in addition to their immediate next action. As we do not input instructions to the policy, we can plan without their specification at test time. Though prior works have used hierarchical structures that generate their own instructions to condition on (Chen et al., 2021c; Hu et al., 2019; Jiang et al., 2019), we surprisingly find that just predicting language instructions is in itself a powerful objective to learn good representations for planning. Teaching agents to output language instructions for completing tasks has two concrete benefits. First, it forces them to learn at a higher level of abstraction where generalization is easier. Second, by outputting multi-step instructions agents explicitly consider the future. Practically, we teach agents to output instructions by adding an auxiliary instruction prediction network to transformer-based policy networks, as in seq2seq translation (Vaswani et al., 2017). Our approach can be interpreted as translating observations or trajectories into instructions.

We test our representation learning method in limited data settings and combinatorially complex enviornments. In the BabyAI benchmark, we attain near the same performance as Chevalier-Boisvert et al. (2019) with only 5% of the data. To our knowledge, we achieve the highest success rate in the Crafting environment from (Chen et al., 2021c), while using fewer demonstrations and no reinforcement learning. We find that in many settings higher performance can be attained by relabeling existing demonstrations with language instructions instead of collecting new ones, creating a new, scalable type of data collection for practitioners. Furthermore, our method is conceptually simple and easy to implement. This work is the first to show that direct representation learning with language can accelerate imitation learning.

To summarize, our contributions are as follows. First, we introduce a method for training transformer based planning networks on paired demonstration and instruction data via an auxiliary language prediction loss. Second, we test our objective in long-horizon planning based environments with limited data and find that it substantially outperforms contemporary approaches. Finally, we analyze the scenarios in which predicting instructions provides fruitful training signal, concluding that instruction modeling is a valuable objective when tasks are sufficiently complex.

## 2  RELATED WORK

Language in the context of policy learning has been heavily studied (Luketina et al., 2019), usually to communicate a task objective. Uniquely, we use natural language instructions to aid in learning via an auxiliary objective. Here we survey the most relevant works to our approach.

**Language Goals.** Language offers a natural medium to communicate goals to intelligent agents. As such, several prior work have focused on learning language goal conditioned policies, particularly for robotics (Nair et al., 2021; Stepputtis et al., 2020; Kanu et al., 2020; Hill et al., 2020; Akakzia et al., 2021; Goyal et al., 2021; Shridhar et al., 2021a), or for games (Chevalier-Boisvert et al., 2019; Chaplot et al., 2018; Hermann et al., 2017). Others in the area of inverse reinforcement learning use language to specify reward functions (Fu et al., 2019; Bahdanau et al., 2018; Williams et al., 2018) or shape them (Mirchandani et al., 2021; Goyal et al., 2019). Unlike these works, we use language instructions that dictate *how* an agent should complete a task instead of language goals that specify *what* the task itself is. Other works, particularly in the visual navigation space, provide agents with instructions similar to those we use, sometimes in addition to language goals. Anderson et al. (2018); Fried et al. (2018); Chen et al. (2019); Krantz et al. (2020); Chen et al. (2021a; 2019) use instructions for visual navigation or drones (Blukis et al., 2019), while Shridhar et al. (2020);

Pashevich et al. (2021); Shridhar et al. (2021b) use instructions for household tasks. Critically unlike our method, these approaches use both language goals and instructions as input, and consequently require the creation of instructions at test-time for every desired task. Other proposed environments (Zhong et al., 2019; Wang & Narasimhan, 2021) assess understanding by prompting agents with necessary information about task dynamics, precluding the removal of text-prompting at test time.

**Language and Hierarchical Learning.** Instead of directly using instructions as policy inputs, other works use language instructions as an intermediary representation for hierarchical policies. Usually, a high-level planner outputs language instructions for a low-level executor to follow. Andreas et al. (2017) and Oh et al. (2017) provide agents with fixed, hand-designed high-level language instructions or policy "sketches". Such approaches require new instruction labels at test-time for every new task unlike our method. Jiang et al. (2019) and Shu et al. (2017) provide interactive language labels to agents to train hierarchical policies with reinforcement learning. In the imitation learning setting, Hu et al. (2019) learn a hierarchical policy using behavior cloning for a strategy game. Unlike the planning problems we consider, their environment has no oracle solution and does not consider generalization to unseen tasks. Most related to our work, Chen et al. (2021c) use latent representations from a learned high-level instruction predictor to aid a low-level policy. However, unlike Chen et al. (2021c), we learn latent representations that can predict instructions, but do not explicitly condition on them at test-time. While hierarchical approaches have shown great promise, the quality of learned policies is inherently limited by the amount of language data available for training. Even with a perfect low-level policy, inaccurate high-level languages commands will yield poor overall performance. This is not an issue for our loss-based approach, as our instruction prediction network can be completely detached from the policy. Additionally, this allows our method to work on a mix of instruction annotated and unannotated data, letting it more easily scale than hierarchical approaches particularly in data-limited scenarios.

**Auxiliary Objectives.** The learning community has extensively studied the use of auxiliary objectives in policy learning. Though to our knowledge no prior works use instructions as an auxiliary objective, auxiliary objectives in general have been found to aid policy learning (Jaderberg et al., 2017). Laskin et al. (2020) and Stooke et al. (2021) demonstrated the success of contrastive auxiliary objectives in robotic reinforcement learning domains. Schwarzer et al. (2020) and Anand et al. (2019) did the same in the Atari game-playing environments. We were inspired by their effectiveness. Additionally, works like Andreas et al. (2018) have previously used language question and answering for representation leaning in visual domains.

**Transformers.** Our approach is based on several innovations involving transformer networks. Vaswani et al. (2017) previously showed state of the art results in machine translation using transformers. While the application of transformers has extended to behavior learning (Zambaldi et al., 2018; Parisotto et al., 2020; Chen et al., 2021b), prior works in the area have not leveraged the transformer decoder. Closest to our domain, Lin et al. (2021) generate captions from video. The architecture of our policy networks take inspiration from recent works adapting transformers to mediums beyond text, namely in vision (Kolesnikov et al., 2021) and offline reinforcement learning (Chen et al., 2021b).

## 3 METHOD

In this section we formally describe the problem of imitation learning with instruction prediction, then describe our implementation for both Markovian and non-Markovian environments.

### 3.1 PROBLEM SETUP

The standard learning from demonstrations setup assumes access to a dataset of expert trajectory sequences containing paired observations and actions $o_1, a_1, o_2, a_2, ..., o_T, a_T$. The goal of imitation learning is to learn a policy $\pi(a_t|\cdot)$ that predicts the correct actions an agent should take. In our work we consider both fully observed (MDP) and partially observed (POMDP) settings. In the partially observed case, policies are given access to previous observations in order to infer state (Kaelbling et al., 1998), and we denote the policy as $\pi(a_t|o_1, ...o_t)$. In the fully-observed setting this is unnecessary, and the policy is simply $\pi(a_t|o_t)$. It is common for policies to be goal conditioned, or even conditioned on language goals as is the case in our experiments. This means they take

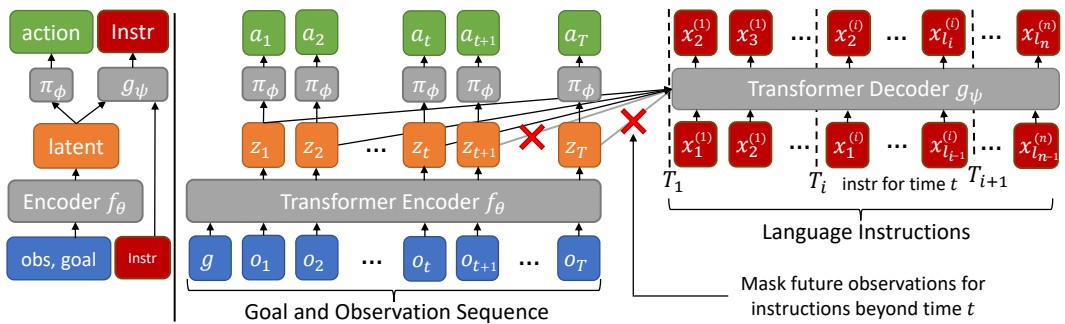

Figure 2: The left diagram depicts the general model architecture used for our approach. Notice how the policy and encoder can be completed separated from the instruction component for mixed-data training or inference. The diagram on the right depicts its implementation for the partial observed environments using a GPT-like transformer encoder. The diagram shows our masking scheme at episode step $t$: latent vectors from beyond time $t$ are masked from the language decoder.

an encoding of the desired task or goal $g$ as input. As our approach works with or without goal conditioning we omit it from the rest of this section for brevity. A standard imitation learning technique is behavior cloning, which in discrete domains maximizes the likelihood of the actions in the dataset using a negative log likelihood objective, $\mathcal{L}_{\text{action}} = -\sum_t \log \pi(a_t|\cdot)$.

In this work, we assume access to oracle language instructions that tell an agent how it should complete a task to provide useful training signal. As mentioned in Section 2, for the purposes of our method we distinguish goals from instructions. Language goals tell an agent *what* to do, whereas language instructions communicate *how* an agent should complete a task. Each trajectory may have several language instructions $x^{(1)}, x^{(2)}, ..., x^{(n)}$ corresponding to different steps in the task. For example, a language instruction like "open the door" only applies to the part of the demonstration before the agent opens the door and after it completes the last instruction. The $i$-th instruction $x^{(i)}$ thus corresponds to an interval $[T_i, T_{i+1})$ where $T_i$ marks the time the instruction was given and $T_{i+1}$ denotes the start of the next instruction. A depiction of an example instruction sequence can be found in Figure 1. While language instructions are an additional data requirement, they can be cheap to obtain, particularly in scenarios where demonstrations are expensive to collect. If one has to collect demonstrations in the real world, providing instructions while they are collected likely only constitutes a marginal increase in effort. Moreover, humans can easily re-label existing demonstrations with instructions. Video data could easily be captioned with voice-over. Similar statements can be made for simulators – if one can code an oracle policy, instructions are likely easy to generate along the way. These modifications can easily be done on simulators with planning stacks, as we do with BabyAI. Moreover, we focus on the data limited regime, where the cost of setting up an environment and collecting more demonstrations is likely be higher than annotating an existing small set of demonstrations. Next, we describe how we train agents to predict language instructions to aid in imitation learning.

## 3.2 INSTRUCTION PREDICTION FOR IMITATION LEARNING

The central hypothesis of this work is that predicting high-level language instructions will force agents to learn representations beneficial for long-horizon planning. In our learning framework, we first construct an observation encoder $f_\theta(o_1, ..., o_t)$ that produces latent representations $z_t$ of the observation(s). Ultimately, it will be trained using both behavior cloning and language modeling losses. As in standard behavior cloning we predict actions using a policy network $\pi_\phi(a_t|z_t)$, which in our case is placed on top of the encoder's latent representation. The same latent representation is also used to predict the current high-level language instruction $x^{(i)}$, where $t \in [T_i, T_{i+1})$. This is accomplished via a language decoder $g_\phi(x^{(i)}|z_t)$. Our general setup is shown in the left half of Figure 2. As is common in natural language processing, we treat each language instruction $x^{(i)}$ as a sequence of multiple text tokens $x_1^{(i)}, x_2^{(i)}, ..., x_{l_i}^{(i)}$ where $l_i$ is the length of the $i$-th instruction. The decoder is trained using the standard language modeling loss. We construct our total imitation learning objective for a given trajectory is as follows

$$\mathcal{L} = -\sum_{t=1}^{T} \log \pi_\phi(a_t|z_t, ..., z_1) - \lambda \sum_{i=1}^{n} \sum_{t=T_i}^{T_{i+1}-1} \sum_{j=1}^{l_i} \log g_\phi(x_j^{(i)}|x_1^{(i)}, ..., x_{j-1}^{(i)}, z_1, ..., z_t) \quad (1)$$

where latent representations $z$ are all produced by the shared encoder $f_\theta$. The first term of the loss is the standard classification loss used for behavior cloning in discrete domains. The second term of the loss corresponds to the negative log-likelihood of the language instructions. We index the language loss by instructions via the first sum. The second summations ensures that we compute the likelihood of instruction $i$ using only observations during or before its execution. The final sum over token log likelihoods is from the standard auto-regressive language modeling framework, where the likelihood of an instruction is the product of the conditional probabilities $p(x^{(i)}) = \prod_{j=1}^{l_i} p(x_j^{(i)}|x_1^{(i)}, ...x_{j-1}^{(i)})$. Finally, $\lambda$ is a weighting coefficient that trades off the importance of instruction prediction and action modeling. During training, we propagate gradients from both behavior cloning and language prediction to the encoder weights $\theta$. In some of our experiments we test additional learning objectives which are also trained on top of the same latent representations $z$ as is standard in the literature (Jaderberg et al., 2017).

Though our method is general to any network architecture, we train transformer based policies since they have been shown to be extremely effective at natural language processing tasks (Vaswani et al., 2017) and carry a good inductive bias for combinatorial planning problems (Zambaldi et al., 2018). For details on the transformer architectures we use, we defer to Kolesnikov et al. (2021); Chen et al. (2021b) and Radford et al. (2018). In the following sections we describe our transformer-based models for both partially observed and fully observed settings.

**Partially-Observed Setting.** For environments that are Partially Observed Markov Decision Processes (POMDPs) or fully observed we use a transformer based sequence model as our policy network, similar to those employed in Chen et al. (2021b). Thus, the model can attempt to infer the true state from all observations. Every observation is preprocessed then fed into a transformer encoder $f_\theta$ to produce latent representations. We operate in the entire sequence at once: $z_1, ...z_T = f_\theta(o_1, ..., o_T)$. Causal masking similar to that in Radford et al. (2018) ensures that at time $t$ the representation $z_t$ only depends on current and previous observations $o_1, ...o_t$. The same policy network $\pi_\phi(a_t|z_t)$ is applied to each latent to produce actions for each timestep. The language decoder $g_\psi$ operates on the same set of latents. Our overall architecture is depicted in Figure 2. Crucially, the decoder employs both causal attention masks to the language inputs and cross attention masks to the latents. Causal-self attention masks on the language inputs enforce the auto-regressive modeling of the instruction tokens. Cross attention masks to the latent representations ensure that predictions for the $i$th instruction cannot attend to latents from timesteps after its execution as is depicted by the red "x"s in Figure 2. This forces language prediction during training to mirror test-time as the agent cannot use the future information to predict what instruction it should execute.

**Fully Observed Setting.** For environments that are Markov Decision Processes (MDPs) or fully observed, we do not need to model the entire sequence of observations and instead use only the most recent observation $o_t$. This corresponds to removing all conditioning on prior latents $z_1, ..., z_{t-1}$ in Equation 1. As such, we employ networks based on the Vision Transformer architecture from Kolesnikov et al. (2021) that predict actions only for a single timestep. Observations are preprocessed into tokens and prepended with a special CLS token: $o_t \rightarrow \text{CLS}, o_{t,1}, o_{t,2}, o_{t,3}, ....$ As we do not input future observations, the transformer encoder uses full unmasked self attention. At the end of the network we take the latent representation corresponding to the CLS token and use it to predict the action $\pi_\phi(a_t|z_{t,\text{CLS}})$. We use all latent tokens to predict the current language instruction with $g_\phi$. A depiction of this architecture would be largely similar to that shown in Figure 2, except the encoder would have only a single observation $o_t$ as input and the decoder would only operate on the current instruction $x^{(i)}$ used for $t \in [T_i, T_{i+1})$. An architecture figure for this model can be found in Appendix C.

## 4 EXPERIMENTS

In this section we detail our experimental setup and empirical results. In particular, we investigate the benefits of instruction modeling for planning in limited data regimes. We seek to answer the following questions: How effective is instruction modeling loss? How does instruction modeling

| Env | Observable | Language | Train Tasks | Test Tasks | Vocab Size | Steps/Instr |
|---|---|---|---|---|---|---|
| BabyAI | Partial | Synthetic | =# Demos | $\infty$ | 45 | 11-12 |
| Crafting | Full | Human | 14 | 35 | 226 | 5-6 |

Table 1: An outline of the differences between the two environments we study.

scale with both data and instruction annotations? What architecture choices are important? And finally, when is instruction modeling a fruitful objective?

## 4.1 ENVIRONMENTS

We test our method on two distinct environments, BabyAI (Chevalier-Boisvert et al., 2019) and the Crafting Environment from Chen et al. (2021c), to evaluate the effectiveness of instruction prediction at enabling long-horizon planning and generalization to unseen tasks. Differences between the two environments are outlined in Table 1. Across both environments, we cover challenges in partial observability and modeling human generated text. Both environments provide temporally delineated coarse high-level instructions.

**BabyAI:** Here, an agent must navigate a partially observable grid-world to complete arbitrarily complex goals specified through procedurally generated language such as locating objects, moving objects, opening locked doors, and more. Goals can also be specified relative to an agents' initial position. Agents are evaluated on their ability to complete unseen missions in unseen environment configurations. We modify the BabyAI demonstration collection agent to output language instructions based on its high level planning logic. Each high level instruction corresponds to multiple low-level actions. The BabyAI environment suite comes with multiple environments of varying difficulty. We focus our experiments on the hardest environment, BossLevel, and use only $1.25\% - 5\%$ of the standard 1 million samples used for agent training with imitation learning. Because of the environment's partial observability and reliance on memory, we employ a transformer sequence model as described in Section 3.2 with the same convolutional network extractor from Chevalier-Boisvert et al. (2019). Goals from the environment are tokenized and fed as additional inputs to the policies. We train all BabyAI models for 1 million steps and evaluate the model that achieves the highest validation action prediction accuracy on five hundred unseen tasks for two seeds.

**Crafting:** This environment from Chen et al. (2021c) tests how well an agent can generalize to new tasks using instructions collected from humans. The original dataset contains around 5.5k trajectories with human instruction labels of which we use 20%-60%. Each task is specified by a specific item the agent should craft, encoded via language. The agent must complete a number of independent steps, collecting and combining resources, to obtain the final item. The tasks vary in difficulty from one-step to five-steps. As this environment is fully observed, we employ the Vision Transformer based model described in Section 3.2. Like in Chen et al. (2021c), we use GloVe embeddings (Pennington et al., 2014) to preprocess both the language goals and instructions, and tokenize the grid as input to our model. For more details on the dataset, we refer the reader to Chen et al. (2021c). We train all Crafting Models for three-hundred thousand steps for four seeds.

For both environments and models the encoder is comprised of four transformer blocks of dimension 128 with two attention heads and the decoder has one transformer block of the same dimension. As both environments use language based goals, we tokenize the goal text, apply embeddings, and append it to the beginning of the observation tokens. Attention then acts jointly over both modalities. More experiments are detailed in Appendix A.

## 4.2 BASELINES

We compare the effectiveness of our instruction modeling auxiliary loss to a number of baselines. The text in parenthesis indicates how we refer to the method in Tables 2, 4, 5, 6, and 7.

1. **Original Architecture (Orig)**: The original state of the art model architectures proposed for each environment in their respective papers. The crafting environment uses a language-instruction hierarchy. In BabyAI, we use convolutions and FiLM layers as in Chevalier-Boisvert et al. (2019).

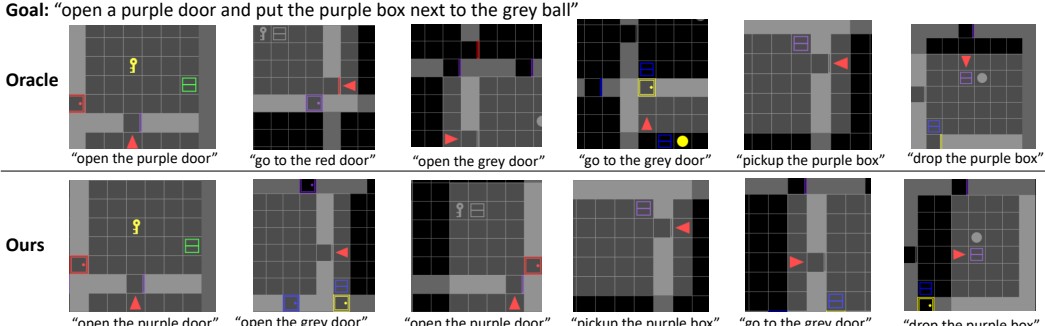

Figure 3: Snapshots of a rollouts from an oracle agent and our trained agents on the same unseen task in BabyAI. Our agent is able to predict instructions, given below each image, with high fidelity. Additionally, we see that our learned agent employs a different exploration strategy, but still completes the task exhibiting strong generalization.

2. **Transformer (Xformer)**: Our transformer based models without any auxiliary objectives to determine the effectiveness of our architectures.

3. **Transformer Hierarchy (Hierarchy)**: A high-level transformer model is trained to output the current instruction that the agent should execute. A low-level executor transformer model is trained to output the actions conditioned on the high-level's language instruction. Prior work has not used hierarchy on observation sequences from partially observed environments like those in BabyAI, and thus we devise our own method. This baseline is designed to compare our auxiliary method to methods in Chen et al. (2021c) and Hu et al. (2019).

4. **Transformer with ATC (ATC)**: Our transformer model with the active temporal contrast (ATC) self-supervised objective proposed in Stooke et al. (2021), which we found to perform better than Laskin et al. (2020) in our environments. This compares representation learning with language to vision based representation learning.

5. **Transformer with Lang (Lang)**: Our transformer based models with the instruction modelling auxiliary objective only.

6. **Transformer with ATC and Lang (Lang + ATC)**: Our transformer based models with both instruction modeling and constrastive auxiliary losses.

### 4.3 How effective is instruction prediction?

Our main experimental results can be found in Table 2, where we compare the performance of all methods on both environments with three differing dataset sizes. We find that for all environments and dataset sizes our instruction modeling objective improves or has no effect in the worst case. In BabyAI, we achieve a 70% success rate on the hardest level with fifty thousand demonstrations and instructions. For comparison, it is worth noting that the original BabyAI implementation (Chevalier-Boisvert et al., 2019) achieved a success rate of 77% with one million demonstrations on a single seed. In the crafting environment, using instruction modeling boosts the success rate by about 5% or more in the 1.1k and 2.2k demonstration setting. To our knowledge our results are state of art in this environment, exceeding the reported 69% success rate on unseen tasks in Chen et al. (2021c).

Visual representation learning was not as fruitful as language based representation learning overall. The combination of ATC and instruction modeling was unfortunately not constructive in all scenarios: it performed better in some instances and worse than just language loss in others. This is consistent with results found in Chen et al. (2021d) that show that observation based auxiliary objectives often yield mixed results in the imitation learning setting. We find that our hierarchical implementations do not perform very well in comparison to plain transformer models. This is likely because with only a few demonstrations high level language policies are likely to output incorrect instructions for unseen tasks leading low-level instruction conditioned policies to output sub-optimal actions. More analysis of the hierarchical baselines is in Appendix B.

| Env | Demos | Orig | Xformer | Hierarchy | ATC | Lang | ATC+Lang |
|---|---|---|---|---|---|---|---|
| BabyAI BossLevel | 50k | 35.3±0.1 | 40.2±2.2 | 36.8±3.5 | 45.8±.6 | **70.3±1.3** | 64.3±0.5 |
| | 25k | 32.3±2.4 | 39.9±0.5 | 37.2±3.0 | 37.1±1.1 | **55.4±7.0** | **56.0±3.0** |
| | 12.5k | 29.9±0.9 | 37.3±0.1 | 36.4±2.6 | 38.4±1.4 | **39.4±1.0** | 38.6±0.6 |
| Crafting | 3.3k | 9.3±0.4 | 74.5±3.3 | 59.9±11 | **75.7±1.0** | 74.5±2.8 | **76.0±2.8** |
| | 2.2k | 4.9±1.0 | 69.4±4.9 | 56.5±9.9 | 73.9±2.1 | 75.2±4.4 | **78.2±4.6** |
| | 1.1k | 1.7±0.8 | 70.1±3.8 | 39.4±3.8 | 70.1±3.7 | **74.8±2.6** | 71.4±2.9 |

Table 2: Success rates (in %) of all methods for varying amounts of demonstration data in both environments. The best method(s) is bolded.

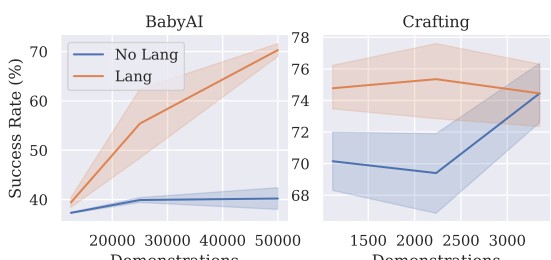

Figure 4: Data scaling with and without instrucutions.

| Model | Base | Lang | ATC |
|---|---|---|---|
| Boisvert et al. | 1052297 | - | - |
| GPT-Like | 661385 | 253056 | 98816 |
| Chen et al. | 1379980 | 218083 | - |
| ViT-like | 579177 | 259712 | 98816 |

Table 3: Model Parameter counts by component. The hierarchical transformer baselines roughly double the parameter counts of their respective models.

## 4.4 HOW DOES INSTRUCTION PREDICTION SCALE WITH MORE DATA AND ANNOTATIONS?

Overall, we find that instruction modeling scales well in the low to medium data regime. With too little data, policies are unlikely to learn good high-level representations that generalize even with the language objective. With a significant amount of data instruction modeling may become unnecessary. Figure 4 depicts how model performance changes with dataset size. In BabyAI, instruction modeling does not appear to significantly help with the smallest number of demonstrations, likely because both the policy and language decoder overfit quickly. However, after that we find that policy performance with language scales almost linearly while policies without language are unable to perform substantially better even with more data. This is not just because training with instructions helps overcome partial observability – we show similar results on a fully observed version of BabyAI in Appendix A.1. The Crafting environment has only fourteen training tasks in comparison to BabyAI's potentially infinite number causing it to require fewer demonstrations to solve. Thus, we observe the opposite problem: instruction modeling helps when the policy is data constrained, and then is neutral when more data is introduced.

A benefit of our loss-based approach is that it can easily be applied to mixed datasets that have only some instruction labels. To additionally study the scaling properties of our language prediction objective, we construct datasets in BabyAI where only half of the trajectories have paired instructions. Results can be found in Table 4. Surprisingly, we find that using half the number of instructions as demonstrations yields significantly more than half of the performance gains from language in the 25k demonstration setting and nearly all the gains in the 50k demonstration setting. As seen in both Tables 4 and 2, *one is better off collecting 12.5k language annotations than collecting an additional 25k demonstrations* in the BabyAI environments. A similar statement can be made in the crafting environment for 1.1k demonstrations. This means that collecting language annotations is a real feasible alternative to collecting more demonstrations.

| % Instr Labels | 0% | 50% | 100% |
|---|---|---|---|
| 50k Demos | 40.2±2.2 | 68.6±1.4 | 70.3±1.3 |
| 25k Demos | 39.9±0.5 | 50.3±1.3 | 55.4±7.0 |

Table 4: We ablate the number of language instruction annotations used in training. 0% means no instructions, 50% means half of the demos have instructions, and 100% means all have instructions. Values are % success rates.

| Model | Success % |
|---|---|
| Lang | 70.3±1.3 |
| Lang No Mask | 50.1±12.1 |

Table 5: Comparison of BabyAI Model on Boss Level with and without masking.

| Level | 50k Demos | | 25k Demos | | 12.5k Demos | |
|---|---|---|---|---|---|---|
| | Xformer | Lang | Xformer | Lang | Xformer | Lang |
| GoTo (Easy) | 88.6±1.8 | **91.0±1.4** | 77.6±2.2 | **82.3±3.3** | **68.9±2.9** | **68.4±2.2** |
| SynthLoc (Med) | 72.5±2.3 | **86.2±1.2** | 60.1±0.5 | **69.4±1.6** | 57.5±0.4 | **58.7±1.2** |
| BossLevel (Hard) | 40.2±2.2 | **70.3±1.3** | 39.9±0.5 | **55.4±7.0** | 37.3±0.1 | **39.4±1.0** |

Table 6: Performance, in percent, of instruction prediction when varying the BabyAI level difficulty.

| Demonstrations | Model | 2 Steps | 3 Steps | 5 Steps |
|---|---|---|---|---|
| | Xformer | **98.1±0.4%** | 66.9±6.2% | **22.1±3.1%** |
| 3.35k | Lang | 96.1±3.2% | **73.0±7.2%** | 19.3±4.4% |
| | Lang+ATC | 97.1±2.2% | **75.1±10.0%** | 13.2±1.7% |
| | Xformer | 89.3±8.6% | 58.4±6.2% | **20.5±1.8%** |
| 2.2k | Lang | **96.1±0.7%** | 73.5±10.7% | 17.3±1.7% |
| | Lang+ATC | 93.9±2.7% | **78.3±13.9%** | **19.8±2.7%** |
| | Xformer | 90.9±4.0% | 58.2±8.5% | 15.0±5.1% |
| 1.1k | Lang | **94.5±1.4%** | **76.0±8.4%** | 13.8±6.0% |
| | Lang+ATC | 89.5±3.2% | 65.2±10.7% | 11.6±2.8% |

Table 7: Difficulty comparison in the crafting environment. Steps indicate the number of steps required for the agent to craft the goal item.

## 4.5 WHAT MODELING DECISIONS ARE IMPORTANT?

In Table 5 we ablate the use of our instruction decoder cross-attention masking. We find that the omission of the masking scheme leads to a 20% drop in performance. Without masking the language decoder has an easier time predicting an instruction as it can attend to observations from after the instruction finished, creating a disparity between train and test time and leading to lower quality representations. Overall, the transformer architecture appears to be critical to high performance, likely because of its good inductive bias for reasoning about objects and their interactions. This is especially evident in the Crafting environment. As stated in Chen et al. (2021c), the imitation learning approaches with the original model were unable to achieve a meaningful success rate on any of the unseen tasks, whereas our baseline transformer achieves a success rate of around 70%. This is better than the reported 69% success rate in Chen et al. (2021c) using all 5.5k trajectories from the dataset and additionally applying reinforcement learning. Our architecture choice is also extremely parameter efficient. As seen in Table 3, our models use significantly fewer parameters than baseline models, even accounting for the additional language decoder.

## 4.6 WHEN IS INSTRUCTION PREDICTION USEFUL?

We hypothesize that instruction prediction is particularly useful for combinatorially complex, long horizon tasks. Many simple tasks, like "open the door" or "grab a cup and put it in the coffee maker" communicate all required steps and consequently stand to gain little from instruction modeling. Conversely, tasks in both environments we study do not communicate all required steps to agents. In the BabyAI environment, exploration is needed to locate objects and implicit tasks arise. For example, agents may need to collect keys to open locked doors. In the crafting environment, goals are a single items that can only be attained through a sequence of steps. Thus, as task horizon and difficulty increase one would expect instruction modeling to be more important. In BabyAI we consider two additional levels – GoTo, which only requires object localization, and SynthLoc which uses a subset of the BossLevel goals. Results in Table 6 indicate that instruction modeling is indeed most important for harder tasks and yields modest gains in the easiest GoTo level. The same trend is true in the Crafting environment as seen in Table 7. All policies are able to craft items requiring two steps tasks with a success rate near or above 90%. However, models with language objectives boast a significant performance boost in the 3-step tasks, from around 58% to closer to 75% in most cases. No agents were able to consistently complete five-step tasks, which is understandable given only one is included in the set of training tasks. The takeaway from these observations is that instructions offer less training signal for combinatorially simple tasks, where reaching the goal requries only a few, obvious logical steps. Thus, we expect instruction modeling to not perform extremely well in benchmarks where instructions can easily be predicted from goals alone. We provide further analysis of this using the ALFRED benchmark (Shridhar et al., 2020) in Appendix A.2. As our

approach works for both procedurally generated and human annotated language, we believe most of the training signal in instructions comes from information about how to solve the task and not the natural structure of human langauge. As task difficulty scales in the future we expect instructions to become a critical modeling component.

## 5 CONCLUSION

We introduce an auxiliary objective that predicts language instructions for imitation learning and associated transformer based architectures. We demonstrate that our language modeling objective consistently improves generalization to unseen tasks with few demonstrations. Moreover, our approach scales efficiently, leading us to conclude that in many settings providing language instructions for demonstrations yields a greater impact than collecting a larger dataset. We further analyze the domains where our method is successful, and make recommendations for when to apply it.

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

# A    ADDITIONAL EXPERIMENTS

## A.1    FULLY-OBSERVED BABYAI

In addition to the standard partially observed BabyAI environment, we created a fully observed version where the agent can view the entire world grid. In this fully observed setting we employ the same model architecture as in the Crafting environment, but with the hyperparameters from BabyAI. Results for a single seed on the BossLevel in the fully observed setting can be found in Table 8, and look largely similar to those of the partially observed BabyAI environment.

| Demonstrations | XFormer | Lang |
|---|---|---|
| 50k | 41.4% | 73.4% |
| 25k | 39.8% | 56.2% |
| 12.5k | 38.4% | 40.2% |

Table 8: Performance, in percent of unseen tasks completed, of instruction prediction loss on a fully observed version of the BabyAI-BossLevel for a single seed.

## A.2    ANALYZING INSTRUCTION INFORMATION AND ALFRED

In section 4.6 we find that instruction prediction is more useful as task difficulty increases. In this section, we try to measure how useful provided instructions are and additionally analyze instruction prediction in the ALFRED visual environment. Instruction prediction is likely to only provide a strong learning signal when they provide information not already available to the agent. Logically this makes sense: if all of the requisite information has been given to the agent in its task, instructions will add nothing new. For example, one goal from the ALFRED environment is "put a watch on the table". If the provided instructions for this task were "pick up the watch", "go to the table", and "put down the watch", the instructions would provide very little useful signal as all of their information was already conveyed by the goal.

By default, ALFRED provides language instructions and goals as inputs to the agent. We remove the instructions from the input so they can be used for our auxiliary instruction prediction objective. Following the methodology and architecture choices of Pashevich et al. (2021), we generate an additional 42K demonstrations in ALFRED and label them with vocabulary from the planner. Overall, we found that instruction prediction had little impact on performance across three seeds as seen in Table 9. Based on these results, we assess why instruction prediction was not fruitful in ALFRED.

We identify two ways in which instructions can provide new information that is useful for training. First, instructions can operate at differing resolutions than the goal. In BabyAI given tasks may involve completing numerous subgoals spanning navigation or object movement, and we provide instructions at this level instead of at the global scale. In the Crafting environment, goals like "make an iron pickaxe" are broken down into instructions like "mine iron ore", "go to the crafting bench". Both scenarios have instructions that give information to the agent at a finer temporal resolution than the goal. If provided goals are too obvious, they cannot practically be broken down at a finer resolution above action-level. The second way instructions can provide useful information is by revealing hidden sub-tasks. For example, instructions in BabyAI may reveal that an agent has to collect a key to unlock a door on the way to its objective. On the other hand, goals from ALFRED tend to be logically simpler as the ALFRED benchmark primarily focuses on visual understanding instead of logical task difficulty. As mentioned in section 4.6, instruction prediction is most effective with harder tasks. Other example tasks in ALFRED include "Put a clean sponge on a metal rack" and "put a cooked potato slice on the counter".

| Success Measure | Xformer | Lang |
|---|---|---|
| Task Success Rate | 28.3±1.0% | 28.5±0.7% |
| Subgoal Success Rate | 36.1±1.0% | 36.0±0.8% |

Table 9: We train models in the ALFRED environment with and without language prediction and evaluate their success on the "seen" validation set.

| Model Inputs | BabyAI, 50K | Crafting, 2.2k | ALFRED, 42K |
|---|---|---|---|
| Text Goal | 86.4% | 43.8% | 96.9% |
| Text Goal and Observation(s) | 92.4% | 49.9% | 99.0% |

Table 10: Instruction prediction accuracies for models trained with and without access to observations. When instructions can be easily predicted without access to observations, they likely provide little additional signal to the agent. The number next to the environment indicates the number of demonstrations used for training. The BabyAI level used was BossLevel, the hardest level.

We can measure how much information instructions are able to provide to an agent by measuring how easy it is to predict them from just the goal. If instructions can easily be predicted from the goal alone, then they are unlikely to provide any additional learning signal to the agent. If the agent can only accurately predict instructions by observing the behavior in the demonstrations and the goal, then instructions prediction is more likely to encourage salient representation learning or logical reasoning about the task. For each benchmark, we take our transformer based architecture for instruction prediction and train only the language head to predict instructions with and without access to observations from the demonstrations. We report the best attained prediction accuracies in Table 10. Our results indicate that the instructions in ALFRED can be nearly predicted perfectly from the text goals (96.9%), indicating that the tasks are too easy and do not require instructions. This result has also been verified by the community. As of writing, one of the top models on the ALFRED leaderboard found here does not even use the language instructions as inputs. Conversely, we see lower accuracy overall and much larger gaps in accuracy for both BabyAI BossLevel (86.4% to 92.4%) and the Crafting environments (43.8% to 49.9%). We hope this result will drive the community to develop more logically challenging benchmarks with complex tasks where the scaling of instruction prediction can be further studied. Additionally, one can begin to assess the effectiveness of instruction prediction before using it by following this methodology.

### A.3 ADDITIONAL BASELINES

We ran a additional baselines in the BabyAI environment.

1. **GPT Enc**: In order to demonstrate that instruction prediction is not just aiding in the agent's understanding of text goals, we construct a baseline that encodes the text goals in BabyAI using a pretrained GPT-2 Model before giving them to the agent. As the text-embeddings have been pretrained, this simulates the case where we have a maximal understanding of the text goal before interacting with the environment.

2. **XFormer AC**: Our original architecture does not use previous actions as input due to the substantial increase in input tokens it causes. This baseline inputs both observation and action sequences into the transformer based model.

3. **Goal Prediction**: Our architecture with language prediction, except instead of predicting the unseen instructions we predict the goal text that is used as input to the policy. This is a type of reconstruction objective in the text regime.

We ran these additional baselines for two seeds. Results for these new baselines and *XFormer* and *Lang* can be found in Table 11. We find that encoding text goals with GPT does not lead to performance gas as large as language prediction. This indicates that our instruction prediction helps with learning good representations for planning, and not just language understanding. The transformer with action inputs (*XFormer AC*) does not perform better than the regular transformer and in fact performs slightly worse, indicating that action inputs are not an important modeling component in the imitation domain and may just make learning harder by adding additional modalities and doubling sequence length. This is different than results found in the Offline RL setting in Chen et al. (2021b), which makes sense as rewards often depend on both states and actions. Moreover, inverse models in the discrete action spaces in BabyAI are relatively easy to learn. Previous works with transformers in imitation (Pashevich et al., 2021) have also found that conditioning on entire action sequences leads to a degradation of performance as policies can more easily overfit.

| Demos | XFormer | XFormer AC | GPT Enc | Goal Pred | Lang |
|---|---|---|---|---|---|
| 50K | $40.2 \pm 2.2$ | $37.4 \pm 0.4$ | $47.6 \pm 0.4$ | $43.5 \pm 1.5$ | $70.3 \pm 1.3$ |
| 25K | $39.9 \pm 0.5$ | $37.0 \pm 0.3$ | $37.6 \pm 0.3$ | $39.6 \pm 1.9$ | $55.4 \pm 7.0$ |

Table 11: Results of additional baselines on the BabyAI Boss Level. The table gives success rates in % on 500 unseen levels.

## B    HIERARCHICAL BASELINES

Here we provide further details on our hierarchical baselines. The two prior works relevant on hierarchical language most relevant to our investigations are Chen et al. (2021c) and Hu et al. (2019). Both of these works learn Markovian, or nearly-Markovian models.

In the crafting environment (Chen et al., 2021c), the authors train a high-level RNN to output the current language instruction. They then condition their low-level policy on the latent representation fed to the RNN that predicts instructions. While they show latent condition to be effective, transformers purposefully avoid encoding entire streams of data into a single vector, and instead operate on a token level. Thus, we found it impractical to attempt this approach with our significantly more effective transformer models.

The environment in Hu et al. (2019) is a partially observed multi-player strategy game. As alluded to in the related work, this environment has multiple viable strategies and is thus distinct from the oracle imitation learning we mostly consider. Though the environment is partially observed, the authors do not train a sequence model. Instead, they concatenate command data from previous time-steps to the model input. As information from the very beginning of the trajectory is necessary for some BabyAI tasks, we found it impractical to scale this concatenation based approach. In Hu et al. (2019), a discriminative high level policy is trained to select an instruction from a fixed set. A low-level is trained with ground-truth human labeled instructions to output actions. In their strategy game, hierarchical approaches perform very well, unlike in our experiments where hierarchical models do not perform the best. One explanation for this comes from the nature of the strategy game environment. The distribution of optimal actions from a given state may be multi-modal, as different strategies may dictate different actions from the same state. Conditioning on an instruction would remove this multi-modality. Below we describe the hierarchical approaches we tried. For all approaches we used the same architectures as detailed in Section 3.

**Fully-Observed Setting.** In the fully observed setting, we adopt a similar strategy to Hu et al. (2019). A high-level policy takes as input the observation $o$ and goal $g$ and predicts the current instruction $x^{(i)}$. We train a low-level policy that predicts actions from the current instruction, goal, and observation $\pi(a_t|o_t, x^{(i)}, g)$. At test time, the high-level auto-regressively generates instructions that are then given to the low level.

**Partially-Observed Setting.** Unfortunately, we find that there is no clear cut way to train a hierarchical model using only transformers where instructions and actions operate at different time scales. Here are the methods we tried:

1. **Sequences for Each Instruction.** We take each trajectory take slices of it up until the completion of each instruction. Our high-level model predicts only the language instruction corresponding to that trajectory slice. This essentially means that the high-level predicts one instruction conditioned on all the history before the instruction. The same sequences are used to train the low-level, conditioned on the single instruction. The low-level policy can be written as $\pi(a_t|o_1, ..., o_t, g, x^{(i)})$. Because training sequence models with only individual losses is very inefficient, we train the low-level model to output the correct actions at all points in time corresponding to the instruction it is conditioned on.

2. **All Instructions**. Instead of conditioning a policy on a single instruction, we train the high level policy to output all of the instructions for the entire task. This is especially challenging at the beginning of an episode when there are few frames. The low-level policy is then conditioned on the entire sequence of instructions and can be written as $\pi(a_t|o_1, ..., o_t, g, x^{(1)}, ..., x^{(n)})$. As this performs relatively well, we hypothesize that the model learns to ignore instructions far in the future when deciding which actions to take at the current timestep.

| Demonstrations | Seq for Each Instr | All Instr | All Instr, Aggressive Mask |
|---|---|---|---|
| 50k | 27.1±2.9% | 36.8±3.5% | 33.3±2.8% |
| 25k | 25.5±3.7% | 37.2±3.0 % | 32.4±2.3 |

Table 12: Results for hierarchical configurations we tried.

3. **All Instructions, Aggressive Mask**. This is the same as the above, except we use an aggressive masking scheme when training the high level that only allows plans to be predicted from observations strictly preceding the time frame of the current instruction.

In Table 12 we give results for each of the sequence-style hierarchical approaches we tried on two seeds. In Table 2 we report the accuracy for the All Instructions method which we found to perform best. Other potential methods could include providing an encoding of the current instruction after each timestep to the transformer. However, such approaches would either employ an RNN or bag-of-words style model to generate the encoding and not be purely transformer based like the rest of our models.

## C ADDITIONAL FIGURES

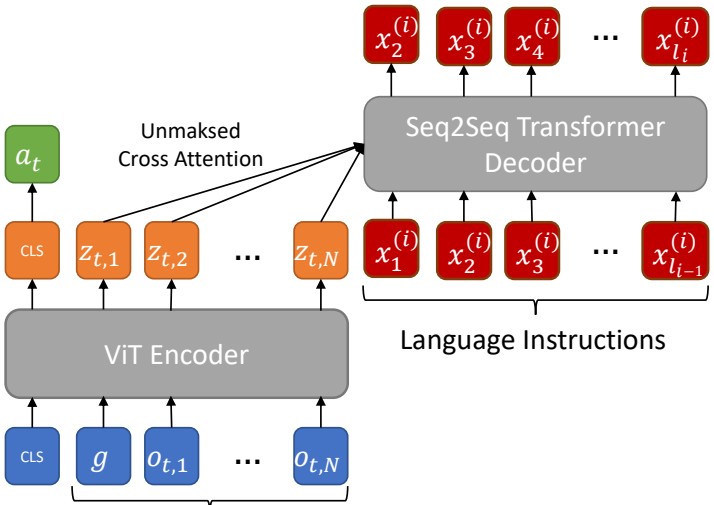

Figure 5: Architecture figure for the fully observed case. Observations are tokenized and then input to a Vision-Transformer like encoder.

## D HYPERPARAMETERS

Here we include all hyper-parameters we used. We used pytorch for our experiments. Our implementation for the partially observed sequence models was based on MinGPT by Andrej Karpathy. Our implementation of the Vision Transformer for fully observed environments was based on Wightman (2019). We determined parameters for ATC by testing different frame skip values in both environments. In the crafting environment we tested using $\lambda$ coefficients of 0.5 and 0.25 and found 0.25 to perform better. We also tested using weight decay and dropout in BabyAI with 50k demonstrations and found it to have no significant impact and thus did not use it for the other experiments. When evaluating models we found those for the Crafting environment to perform better in Pytorch "train" mode, meaning with dropout on, while those for BabyAI worked better in "eval" mode. We ran our experiments on NVIDIA GTX 1080 Ti GPUs. In BabyAI we train all models for two seeds and in the crafting environment we train all models for four. This was done because BabyAI experiments could take upwards of 48 hours to train.

| Hyperparameter | BabyAI | Crafting |
|---|---|---|
| Encoder Blocks | 4 | 4 |
| Decoder Blocks | 1 | 1 |
| Embedding Dim | 128 | 128 |
| MLP Size | 256 | 256 |
| Dropout | 0 | 0.1 |
| policy $\pi_\phi$ | Dense Layer | Dense Layer |
| Batch Size | 32 | 64 |
| Training Steps | 1 million | 300k |
| Optimizer | Adam | AdamW |
| Optimizer Epsilon | $1 \times 10^{-8}$ | $1 \times 10^{-8}$ |
| Learning Rate | 0.0001 | 0.0001 |
| Weight Decay | 0.0 | 0.05 |
| Grad Norm Clip | N/A | 1 |
| $\lambda_{\text{lang}}$ | 0.7 | 0.25 |
| $\lambda_{\text{ATC}}$ | 0.7 | 0.25 |
| EMA $\tau$ | 0.01 | 0.01 |
| EMA update freq | 1 | 1 |
| ATC Frame Skip | 3 | 1 |

Table 13: Hyperparameters

