# OpenReview forum: "Improving Long-Horizon Imitation Through Language Prediction"
_ICLR.cc/2022/Conference — ICLR 2022 Submitted_

### Official Review · Reviewer_hiJp · 2021-10-31

**Correctness:** 2
**Technical Novelty And Significance:** 3
**Empirical Novelty And Significance:** 1
**Recommendation:** 5
**Confidence:** 4

**Main Review:**

I think the core method of this work is quite simple (which is not inherently bad and I would even argue is inherently good). The authors add a auto-encoding objective to a transformer policy network, where during each step, an additional decoder ingests 1) the instruction for that step and 2) the observation encoder output for all previous steps and recovers the instruction for that step. This reconstruction loss is weighted with the usual policy loss and jointly optimized. On the BabyAI and Crafter experiments, this method achieve strong gains using fewer demonstrations (for imitation learning) and converges faster and to higher performance. I think it is quite clear that this method is very helpful for the tasks studied in this work.

What I find lacking is how this comparison compares to the general claim that instruction modelling is useful in the more general sense (although I hope it is and would like to believe it is). I think what would significantly improve this paper is to study instructions with more complex behaviour which truly suffers from the lack-of-annotation problem. There are a number of such environments such as Touchdown (navigation with streetview panoramas, https://arxiv.org/abs/1811.12354) and DRIF (quadcopter control, https://arxiv.org/pdf/1910.09664.pdf), where due to the complexity of the trajectories and observation space it is prohibitively expensive to annotate language instructions. This would evaluate this technique on complex language instructions. Another way to go is to do more complex grid worlds with high stochasticity and more complex dynamics that actually require detailed understanding of the language because test language-entity associations are unseen and must be constructed from language. Examples here include RTFM (7+ hop reference resolution, https://arxiv.org/abs/1910.08210) and Messenger (NL instructions, small training distribution, https://arxiv.org/abs/2101.07393). These are also good related reference for the paper, which this draft doesn't include - some of the references this work does include would also be good test beds - because they also contain more complex grounding challenges due to visual observation complexity.

All of these previous works include dynamics and language instructions more complex that the ones studied in this paper. In particular, due to construction, once instructions are broken down into atomic steps, they don't even require sequential understanding. For example the instruction "go to the red door" is perfectly recovered with the binary features (goto, red, door). As a result, the experiments leave many questions unanswered, such as

1) does this result generalize to complex environments where language-entity and language-dynamics associations are more difficult due to complex entity behaviour or complex observation?
2) does this result generalize to complex language instructions where the instruction is more difficult to model (and therefore to understand)?

Update because I looked at appendix:
The negative results from ALFRED further raises my suspicions that these results may not generalize. For the results in the Appendix: are the authors claiming that one can do ALFRED without visual observations? My understanding is that the ALFRED work shows exactly that you cannot do the task without visual observations (it is one of the ablations in their paper). Perhaps I am misunderstanding this result. Can the authors please comment on this?

I would imagine that the result shown here is because the model is incapable of modelling the instructions, and therefore the reconstruction loss is not helpful. In particular, I would think that there is a high correlation between being able to do the task and being able to reconstruct instructions from the bottleneck representation. If the instructions are very complex to understand, then both are hard and the reconstruction loss is not helpful. One way to test this hypothesis is to plot the instruction recovery accuracy instead of loss (the authors show the latter but not the former).

**Summary Of The Paper:**

This work proposes using instruction modelling as an auxiliary objective to improve long-term planning. On BabyAI and Crafter, this method demonstrates significant improvement on longer hop levels (e.g. 3 hop BabyAI).

**Summary Of The Review:**

Strong results that show instruction modelling help policy learning on two simpler tasks. Evaluation leaves much to be desired in terms of substantiating main claims.

---

> ### Author Response · Authors · 2021-11-11
> **Response to Reviewer hiJp Part 2/2**
>
> 3.*The suggested grid-world environments.*
>
> We again thank the reviewer for bringing these papers to our attention as they are indeed very interesting. We have now cited these as well. Both RTFM and MESSENGER focus on an agent's ability to read or understand language descriptions and use them to identify the dynamics of the task. In RTFM documents contain some instructions, but mostly help provide information on dynamics and not “how” an agent should complete a task. In Messenger, the agent needs the information in the text descriptions to determine the goal. Thus, the descriptions can’t be used for instruction prediction. For language prediction to be applicable to either of these environments, we would require that the tasks be reasonably solvable without the documents at test time. Moreover, both environments are reinforcement learning based instead of focusing on imitation.
>
> 4.*Do our results generalize to settings where instructions are more difficult to model?*
>
> Note that the Crafting environment uses human-annotated instructions that present the usual challenges of natural language: synonyms, complex syntax, etc. Our results indicate that even with more difficult to model language input language prediction still improves performance, showing an up to 15% gain on 3-step tasks as seen in Table 7. We have also made changes to place greater emphasis on “instruction” modeling  versus “language” modeling in the introduction. Finally, we have added an additional baseline in appendix A.2 that uses pretrained GPT-2 weights to encode the environment’s goal text, which does not perform substantially better than the default transformer model. This indicates that the challenges in the presented environments are not in language understanding, but rather in planning.
>
> 5.*The ALFRED Results*
>
> We do believe there may be a mis-understanding in the interpretation of the ALFRED results. We have entirely re-vamped this section of the appendix in the following ways:
>
> * Added experimental comparisons with the other environments
> * Changed from showing loss to showing the highest attained recovery accuracy, as suggested in the review.
> * Improved the writing to make it clearer.
>
> Here is an in-depth explanation to hopefully clarify the purpose of our experiments:
>
> In the ALFRED challenge, agents are given natural language goals and natural language instructions for how to complete them. The Episodic Transformer paper (Pashevich et al. 2021) demonstrated that better performance could be achieved by learning a translation model from natural language to synthetic PDDL derived language and inputting that to the model. To make the ALFRED task applicable to our problem setup, we created a variant of the benchmark where only task goals were given as input to the agent and instructions could be used for language prediction. Our experiments with this setup and the same vocabulary from ET yielded no significant gain from language prediction. We concluded that this meant that the language instructions used in ET provided little additional learning signal about how the task should actually be completed.
>
> To test this hypothesis, we try to predict the language instructions from only the language goal. If instructions can be predicted from just the goal, then no representation learning or logical reasoning about the environment would be encouraged by language prediction. If we need observations to predict instructions, the one might posit some representation learning happens at the observation level when instructions are predicted.
>
> Overall, we find that in the ALFRED environment, language instructions can be predicted with almost perfect accuracy (97% vs 99%) without any visual observations from the environment. This indicates that predicting the instructions is unlikely to be a useful objective in the benchmark. In BabyAI and Crafting, we find that observations are necessary to predict the instructions. Prediction accuracy increases from 85% to 92% in the presence of observations in BabyAI BossLevel and from 44% to 49.9% in Crafting. This makes sense -- when instructions include hidden subtasks like “get the door key” or operate at different time scales, they might not be easily predicted from tasks like “go to bedroom”. Conversely, they are also more likely to be useful to the agent.
>
> Our results on ALFRED have also been publicly verified to an extent. One of the top submissions on the ALFRED leaderboard does not use the language instructions at all, and only inputs goals (https://leaderboard.allenai.org/alfred/submission/c2v3806nv9gjbp2idigg), indicating that instruction modeling may not be able to provide performance gains in ALFRED.
>
> We hypothesize that this is because a lot of the tasks in ALFRED are logically too simple for instructions to be useful. For example, tasks like “put the watch on the table” communicate almost all the steps needed.

---

> > ### Comment · Reviewer_hiJp · 2021-11-19
> > **Response**
> >
> > First of all, I'd like to thank the authors for the detailed clarification RE ALFRED, I think this is a great addition to the manuscript. To make sure, when you say "goal" do you mean the NL instruction? and then when you say "instruction" you mean the synthetic PDDL-derived language? Do you think for the presented tasks, the one can also predict the "instruction" with very high accuracy given the "goal"? My impression is yes, if you train a model with goal-instruction pairs analogous to the follow-up ALFRED experiment. Having this result would also resolve your new hypothesis of whether the ability to predict instructions affect whether these language yield additional learning signals.
> >
> > > This indicates that the challenges in the presented environments are not in language understanding, but rather in planning.
> >
> > I think this is exactly right - the primary challenge in the presented environments have to do with planning as opposed to language. Hence I am curious whether this approach generalizes to tasks where language understanding poses a larger challenge (e.g. Touchdown). However for these langauge-rich environments, the authors seem to categorize the language under "instructions" as oppose to "goals" and therefore excluded from this study.
> >
> > My impression is that when the goal is sufficiently complicated, one must use increasingly complex language to describe it. Consider two scenarios. In the first, all blue keys open blue doors. Hence, I can infer from the "goal" of "open the blue door" that I need to first "get the blue key". In the second, there are 5 blue keys instead of 1 blue key, and only one of them opens the blue door. Now I cannot reliable solve the goal, unless the goal giver helps me disambiguate which 1 of the 5 blue keys I need. I believe Touchdown is exactly this setting, but instead of disambiguating between blue keys we are disambiguating between entities in real scenes. In other words, I do not believe the distinction between "instructions" and "goals" is as clear cut as the authors make it out to be in these other tasks with richer language.

---

> > > ### Author Response · Authors · 2021-11-20
> > > **Thank you for looking at our revised manuscript!**
> > >
> > > Thank you for looking at our revised manuscript! We are happy that you find the new section on ALFRED to be useful.
> > >
> > > To clarify the question asked, we use the same setup for goals and instructions found in Episodic Transformer, Pashevich, Schmid, Sun 2021. (https://arxiv.org/abs/2105.06453). “Goal” refers to the natural language description of what to do, eg. “Put two vases on a cabinet”. “Instruction” refers to the step by step directions, eg “goto to vase”, “pick up the vase”, “goto the table”. We use the same architecture, hyperparameters, and vocabulary as Pashevich et al. In order to use extra data, Pashevich generate more demonstrations in ALFRED, label them with PDDL based language instructions, and learn a model that translates NL instructions into the same space as the PDDL based language instructions. We test our model with the same inputs as  Pashevich et al. thus the PDDL based language instructions.
> > >
> > > I believe in the ALFRED section we do exactly what you say, we predict the “instructions” with high accuracy given just the “goal” in ALFRED. See table 10.
> > >
> > > It is true that the distinction between goals and instructions can be a bit convoluted, however, we hope that the Appendix section on instruction information clarifies that our method can provide a boost when instructions are text that provide additional information on how to complete a task. For example, they could just be finer grained instructions that one does not want to have to provide at test time.

---

> ### Author Response · Authors · 2021-11-11
> **Response to Reviewer hiJp Part 1/2**
>
> We would like to thank the reviewer for their in-depth comments and for finding the simplicity of our approach “inherently good”. We would first like to begin by clarifying our problem setup, and then will address items in the review point by point.
>
> A laudable goal in robotics and AI is to build agents that can complete long-horizon and combinatorially complex tasks without excessive prompting. Imagine that we want a robot in a bedroom to go get a snack from a refrigerator. We would only want to tell the robot “get me a snack” (the goal), and not have to provide it with details like “go to the door, open it, leave the room, ... , go the refrigerator, open the refrigerator, …” (the instructions). Concretely, we want to provide a goal and not explain how to achieve it at test-time. We develop a method that can use instructions at train-time to see large performance gains, but not need them during inference. With this in mind, we will address individual concerns brought up in the review.
>
> 1.*The usefulness of instruction modeling in more settings*
>
> One particular challenge in the problem setup mentioned above is that most environments that use language instructions suffer from one or both of the following problems:
>
> 1. The environments use language instructions as inputs. As argued above, we would like to not have to provide agents with instructions at test time. For benchmarks, particularly in visual instruction following like R2R, Touchdown, and DIRF, instruction following is the goal. AFLRED also uses language instructions as inputs by default. These benchmarks test whether agents can follow pre-specified steps, and thus will require instructions at test time.
>
> 2. The tasks are not complex enough, or the instructions are not fine-grained enough. As shown in Tables 6, 7 language prediction scales with task difficulty and in the presence of unspecified steps. For many existing benchmarks, steps are fully specified in the model input (instructions), or the instructions are too easily inferred from just the task text to provide a useful signal in language prediction. This is discussed in more detail in the updated Appendix A.2 section on instruction information and ALFRED.
> We do believe that in goal based long-horizon and combinatorially environments instruction prediction can lead to better performance. Many existing benchmarks that increase difficulty on the visual axis do not also push difficulty on the logical reasoning axis, making instruction prediction less fruitful. We believe it's more reasonable to compare our environments to those used in related works in language based hierarchical planning instead of those used in visual instruction following. A perhaps undervalued aspect of what we have shown is that in many cases hierarchy is not needed to achieve the gains from language shown in other works.
>
> 2.*The Suggested Visual Environments*
>
> We thank the reviewer for bringing Touchdown and DIRF to our attention. We have added references to these in the paper. We had previously examined these environments, and as mentioned above found them by default in-applicable to our setup because they depend on instructions as inputs. If instructions are required as inputs, we have nothing to predict with the language decoder.

---

> > ### Comment · Reviewer_hiJp · 2021-11-19
> > **Instructions vs goal**
> >
> > While I agree with your general sentiment that ideally we should provide goals and not instructions, the "goals" studied in this work (e.g. open the blue door) are similar in terms of complexity to one "instruction" in some of the other works. This is not to mention that the worlds studied in this work are also much simpler than those found in visual instruction following.

---

### Official Review · Reviewer_z2VY · 2021-11-03

**Correctness:** 3
**Technical Novelty And Significance:** 1
**Empirical Novelty And Significance:** 2
**Recommendation:** 5
**Confidence:** 5

**Main Review:**

The novelty of this paper is limited. The authors use language as an auxiliary task to improve planning accuracy. Such a method has no technical contribution. I would like to see the summarization of the technical contributions from the authors in their feedback.

The experimental results are not very interesting. It is not surprising that using extra supervision will improve the results.

Could the authors explain the difference between the claimed "non-Markovian" and the commonly used "partially observable Markov decision process (POMDP)"? Why do the authors use "non-Markovian" instead of "partially observable Markov decision process (POMDP)" in the paper?

In the Crafting environment, the results of "Lang" are comparable with the results of "Xformer" in Table 2. Is the model of "Lang" also larger than "Xformer" as it has the extra language encoder and decoder? Could the authors give any explanation of why using extra supervision generates comparable results?


**Summary Of The Paper:**

In this work, the authors explore the use of language as auxiliary supervision for long-horizon imitation learning tasks. The authors show that using instruction modeling is able to improve the performance in planning environments when training with a limited number of
demonstrations on the BabyAI and Crafter environments. They further show that instruction modeling is most important for tasks that require complex reasoning in the experiments.

**Summary Of The Review:**

After reading the whole paper, I didn't find any interesting points that make me feel excited about this paper. I think the authors did good experiments to verify their thoughts, but this paper has no technical contribution and the results are not surprisingly good enough.
This paper is easy to read and has no significant fault.

---

> ### Author Response · Authors · 2021-11-11
> **Response to Reviewer z2VY Part 1/2**
>
> We thank the reviewer for their in-depth comments. As requested, we will begin with a summary of what we believe are the core contributions of our work:
>
> **Contribution Summary**
>
> *Algorithmic*: We do pull from many prior works, but believe the following aspects of our work are unique:
>
> 1. Using language instructions for representation learning in control.
>
> 2. Our architecture is based on several prior works, but combines them in a unique way, using language decoders on top of visual or control based encoders.
>
> 3. We develop a unique causal masking scheme for instruction modeling and demonstrate its importance (Table 5).
>
> 4. Crucially, we use language instructions in a non-hierarchical manner unlike many other works (Chen et al. 2020, ), and do not use any language at test-time. To our knowledge, there are no other representation learning approaches for imitation with instructions that are non-hierarchical.
>
> It’s worth noting that in line with the comments of reviewer hiJp, we believe the simplicity of our approach is inherently good. It means it makes it easy to reproduce or use plug in to existing systems. Simple tricks have also been shown to make huge impacts in the field. For example, image augmentations were by no means new, but by showing they were effective in pixel based control, RAD (https://arxiv.org/abs/2004.14990) had a huge impact in RL. The paper went on to earn a spotlight presentation at NeurIPS.
>
> *Empirical*
>
> 1. The most sample-efficient results on BabyAI with imitation
>
> 2. SOTA results on the crafting benchmark
>
> *Impact*
>
> 1. We demonstrate that our non-hierarchical method can achieve similar or better performance than prior hierarchical language methods devised for imitation learning. Many hierarchical works in this area (Chen et al 2021, Hu et al. 2019) compare only to non-hierarchical methods that don’t use instructions. We believe our work will be impactful because it shows the community how strong non-hierarchical approaches can be.
>
> 2. We show that collecting alternative types of data for quality representation learning, specifically language instructions, can be more useful than collecting more demonstrations themselves in many cases. This is impactful because it shows practitioners that there are potentially cheaper means of obtaining higher performance than previously thought.
>
> 3. We move away from using instructions at test-time, which is also unique in comparison to lots of works in the visual-language space.

---

> ### Author Response · Authors · 2021-11-11
> **Response to Reviewer z2VY Part 2/2**
>
>
> **Point by Point Remarks**
>
> *“It is not surprising that using extra supervision will improve the results”*:
>
> Multi-task learning is not always constructive (https://arxiv.org/pdf/1905.07553.pdf), and is sometimes actually detrimental to performance. We would argue that language supervision being effective for control is not entirely obvious. What we believe to be most interesting about our results is that they show that the representations encouraged by language prediction are actually more impactful than adding additional demonstrations. Table 4 shows that collecting instruction annotations can be more effective than collecting more demos. In particular, you get 15% higher performance in BabyAI with 25K Demos and 25K language annotations than you do with 50K demos. This is surprising because optimizing an auxiliary objective for representation learning is actually more effective than collecting data for the final objective.
>
> *"non-Markovian"  vs. POMDP*:
>
> We originally used the non-Markovian for brevity to indicate that the environment did not follow the markov property. We have updated the text to state POMDP and MDP where applicable.
>
> *Crafting Environment Results*:
>
> It is true that the performance gap is smaller in the Crafting environment than in BabyAI. One key difference between the two environments is that the crafting environment only has 14 training tasks, while BabyAI has a different task in every demonstration. A more nuanced explanation of the scaling properties in the crafting environment can be found in section 4.4, where we discuss how language prediction is more useful when policies are data constrained. However, we still see a gain. In the 1.1k and 2.2k demo settings, XFormer has around a 70% success rate while methods with language modeling losses achieve around 75% or higher. In section 4.6, we breakdown the success rates in the crafting environment by the number of steps to complete an unseen task. Note that the numbers for the 2-step tasks are very similar between all models. However, the models with language experience around a 15% gain in performance on the 3-step tasks in comparison to the Xformer with 1.1k and 2.2k demos. These results support our hypothesis even more -- language prediction is important for learning longer horizon tasks, and less important for shorter horizons. Note that performance in the 5-step tasks is a bit of a wash because there is only one five-step task included in the training dataset, making generalization very tough.

---

### Official Review · Reviewer_cgzc · 2021-11-03

**Correctness:** 3
**Technical Novelty And Significance:** 2
**Empirical Novelty And Significance:** 3
**Recommendation:** 6
**Confidence:** 4

**Main Review:**

I like the overall problem and conclusion that predicting detailed instructions can improve performance using much smaller data than without this auxiliary objective. I think the perspective that labeling subtasks with language instructions is more sample efficient than collecting more demonstrations is also useful. I have some questions about the approach.

1- Could you explain if "instruction as a text" or just "objects and actions in instructions" are more important? Based on Figure-3, it seems that each instruction includes an object and an action. It would clarify if your auxiliary objective is helping just the object detection, subtask understanding, even using OOD information that wouldn't otherwise be in the goal (such as a grey door is never seen in a goal but it is a part of navigating the environment), or if actual text is also important.

2- What is the main bottleneck that instruction generation is solving? Similar to my above question, it could be object detection or subtask understanding but might also be that understanding a textual goal is difficult. A pre-trained language model for goal encoding would help answer this.

3- To understand if detailed instructions are really crucial for instruction generation, an auxiliary loss baseline where goal is predicted at each time step is needed. It would help understand if text generation or detailed instructions are the most important; even for POMDPs, partially generated goals might be helpful.

4- You use only observations and goal as input to the transformer encoder but DecisionTransformer [1] showed that using action and reward was crucial for success. I think a similar baseline with and without language prediction is needed.

5- In Figure-2 and Figure-6, decoder outputs should be shifted by 1 but right now the same input is outputted at each step.

6- In Table-1, you put "=# Demos", was this intentional or a placeholder for an exact number?

7- Some grammatical errors:
* In Section 3.1, "works with our without" should be "works with or without".
* The same section, "langauge" should be "language".
* Section 3.2, "depcited" should be "depicted".
* The same section, "except it the encoder" should be "except the encoder".
* Section 4.1, "to an agents initial" should be "to an agents' initial".
* Section 4.2, "Prior work has not use" should be "Prior work has not used".
* Appendix A.2, "AFLRED" should be "ALFRED".
* Appendix B, "to select a instruction" should be "to select an instruction".

[1] Decision Transformer: Reinforcement Learning via Sequence Modeling. Lili Chen, Kevin Lu, Aravind Rajeswaran, Kimin Lee, Aditya Grover, Michael Laskin, Pieter Abbeel, Aravind Srinivas, Igor Mordatch.

**Summary Of The Paper:**

This paper investigates the usage of language prediction for imitation learning. The authors distinguish between a goal (a short description of the task) and an instruction (a detailed description of subtasks that span a sequence of observations). They show that predicting instructions at each step as an auxiliary loss improves the performance using lesser number of demonstrations. They experiment with minigrid and crafting environments to show the efficacy of the proposed approach.

**Summary Of The Review:**

I think instruction generation as an auxiliary loss is an interesting approach and presents a new perspective that labeling subtasks with instructions is better than just collecting more demonstrations. I also think that some more baselines and clarifications are needed.

---

> ### Author Response · Authors · 2021-11-11
> **Response to Reviewer cgzc**
>
> We thank the reviewers for their detailed comments and for identifying the “new perspective” we hoped to share on data collection for imitation learning. The reviewer detailed their primary concerns point by point, which we will address below.
>
> 1.*Instructions as Text vs objects and actions in instructions*
>
> We posit that instructions provide a useful signal to agents because they act as a “long-horizon” action or subgoal label. To answer your question directly, we believe our experiments show that ‘actions in instructions’ is more important than the fact that instructions are language because we show that our approach works on both synthetically generated (closer to just actions in instructions) and human labeled data. Both environments share ‘action in instruction’, and we see performance gains in both. The fact that instructions are generated as text in BabyAI may not be important, but was chosen for consistency with the existing BabyAI benchmark. We have mentioned this in the main text in section 4.6. I believe this comment about having instructions as text also ties into the next bullet point, for which we ran an additional baseline showing that text understanding is not the performance bottleneck in babyAI.
>
> 2.*Are instructions helping with text understanding?*
>
> To answer this question, we trained a model where the text goal is encoded using GPT2, and include results and discussion in appendix A.3. We find that just adding in the text-understanding of GPT-2 does not significantly improve planning. This indicates that instructions are helping with high-level reasoning and not just understanding text.
>
> 3.*"an auxiliary loss baseline where goal is predicted at each time step”*
>
> Could the reviewer please clarify what they mean by this baseline? As noted in the review we distinguish between “goals” and “instructions”. As goals are an input to the policy, we expect predicting them to be rather easy and unlikely to provide a training signal.
>
> 4.*Actions as input to the decision transformer*
>
> We ran this baseline and have included it in appendix A.3. Note that it is not applicable to the Crafting environment, as the model for the crafting environment does not take in observation sequences. We find that inputting actions does not increase performance, and is actually slightly detrimental to performance. This is likely because adding in actions doubles the horizon of the policy and introduces another modality. In the Offline RL setting used in Decision Transformer, reward prediction is necessary, which depends on both actions and states. In imitation learning, we have no output dependence on previous actions. Other papers (Episodic Transformer, Pashevich et al 2021) have found that adding a large action history can actually hurt imitation learning performance by making it easier to overfit.
>
> 5.*Decoder text should be shifted*
>
> This has been fixed!
>
> 6.*What does # = Demos mean?*
>
> This was meant to convey the fact that tasks in BabyAI are randomly generated, and thus each demonstration is of a unique text goal and initial state pairing. Thus, a dataset of 1000 demonstrations contains examples of 1000 tasks. So number of training tasks = number of demonstrations.
>
> 7.*Grammar Errors*
>
> These have been fixed, thank you for pointing them out!
>
> In the light of the above clarifications, we would like to ask if you are willing to increase your score assuming we have addressed your concerns. Otherwise, please let us know if you have additional questions.

---

> > ### Comment · Reviewer_cgzc · 2021-11-12
> > **Thanks for the prompt response.**
> >
> > Thanks for additional experiments and analysis. I think GPT-2 results somewhat clarifies my understanding of what the proposed model is solving.
> >
> > 1. I think the explanation for why your model didn't work for ALFRED makes sense to me. But, I am still a bit skeptical if this would generalize to more complicated benchmarks such as RxR [2] dataset which is closer to your assumptions where room-level instructions are needed for language prediction to be useful. Indeed, in RxR, instructions and observations are already temporally aligned which can be used to test your model without additional data collection.
> >     * Could you discuss if language prediction would also be useful here? For example, take the example in Figure-1 in the [2], and let's describe a much less detailed goal such as "Find the shower area and stop". Do you think your model would also work by just using this high-level instruction without using any of the detailed instructions as input? It is not clear to me if the model would actually solve this problem by just language prediction as it would require exploring the whole house to locate the area.
> >     * If you need more detailed instructions to solve the problem, how useful would language prediction be in that case?
> >
> > 2. *As goals are an input to the policy, we expect predicting them to be rather easy and unlikely to provide a training signal.* I think this is not necessarily true as all autoencoders are trained to reconstruct the same input signal. Since you don't output a hidden vector for the position of the goal (in Figure-2 right), the encoder would need to remember the goal for the decoder to reconstruct it correctly.
> >
> > 3. Thanks for the decision transformer experiment. Could you explain what you mean by *reward prediction is necessary* ? In Decision Transformer, there is no reward prediction but it is merely an input to the model.
> >
> >
> > [2] Room-Across-Room: Multilingual Vision-and-Language Navigation with Dense Spatiotemporal Grounding. Alexander Ku, Peter Anderson, Roma Patel, Eugene Ie, Jason Baldridge.

---

> > > ### Author Response · Authors · 2021-11-12
> > > **Thank you for the response!**
> > >
> > > Thank you for the prompt response! Here are clarifications to your points:
> > >
> > > 1. *The RxR benchmark and room-level instructions*: we are familiar with the R*R family of datasets, and did look at using them. Note the distinction between goals and instructions we use in our work, which I believe may have caused a slight mis-understanding. To be clear, our setup assumes that the inputs to the policies, including any language, is different than the language used for instruction prediction.
> > >
> > > * As mentioned in some of our responses to other reviewers, we strive to learning policies that do not require laborious prompting at test time. In R2R, R4R, and RxR, the model uses language instructions *as the goal*.  We actually cite R2R in our introduction as doing this. The text input provided to the policies in these benchmarks is something like: "we start in the living room, facing a couch. Turn around and go through the door ..., finally go to the shower and stop". These benchmarks are designed to assess an agents ability to understand and follow instructions, and require them at test-time. Our setup is about achieving goals. As shown in figure 2, instructions can be completely detached from the agent in our method. This makes it difficult to test our method on these benchmarks as they assume instructions are input directly to the policy. If we remove the provided instructions from the policy input to use them for language prediction, the agent would have no way of determining what it should do!
> > >
> > > * Our method would be applicable in a scenario where we have the following breakdown of information. Goal: "go to the shower", instructions: the detailed instructions provided in figure 1 of [2]. The direct policy inputs would be ("go to the shower", o_1, o_2,  ...., o_t), outputs would be the actions (a_1, ..., a_t) and the detailed instructions would be used in the decoder (x_1, ... x_L) in our figure 2. In this scenario, language prediction would perhaps encourage the policy to learn representations that latch on to key aspects of the environment given in the detailed instructions, but would not require them at test time. As mentioned above this breakdown is not available in RxR. Even if this breakdown were available, the problems presented in RxR are just navigation, and as we show in Tables 6 and 7, logical difficulty and not visual difficulty are likely where instruction prediction will excel. For example, when making coffee humans think about steps like "get a cup, get beans, fill the coffee maker, etc.", but we don't really think about pure navigation in steps. In the BabyAI GoTo level, which is only navigation, performance gains drop from 30% to only 3% versus BossLevel. This is consistent with our argument that instruction prediction helps with logical difficulty.
> > >
> > > *  *Would only using "find the shower area" as an instruction be enough*. I believe there was some slight misunderstanding, remember in our setup that goals are different than the instructions used for language prediction. Nonetheless, the new experiments in Appendix A.2 show that language prediction is more helpful when instructions contain a larger amount of information. It seems unlikely that any gains would come from such instructions, as is empirically seen in AFLRED ,
> > >
> > > 2. *Reconstruction Objective*: It is true that auto-encoders use reconstruction as an objective, but they also require some type of regularization that enforces simplicity or a dimensionality constraint on the learned latent representation. The transformer architecture does not have either of these bottlenecks, which is actually one reason why it performed better on translation than LSTM encoder-decoders (Attention is All You Need, Vaswani et al 2017). Second, the new results in Appendix A.2 show that it language prediction is useful when it's hard to predict the instructions from just the goal. A pure reconstruction of the goal text is likely to be very easy to predict, so we don't believe it would help. Nonetheless, if we get the time we can try to run a baseline like this and report results.
> > >
> > > 3. *Decision transformer (DT)*: Sorry, we should have been more careful about our statements. DT conditions on reward, and actions usually have some impact on reward r(s,a) , thus we think that actions have a greater impact there. For the BabyAI and Crafting environments, we also found that perfect inverse models were extremely easy to learn with only a few thousand samples given that the action spaces are discrete, making action inputs seem obsolete. Given both of these, we think it is understandable that inputting actions did not lead to an increase in performance in the imitation setting. In Episodic Transformer (Paschevich et al.) the  authors also found that conditioning on entire action sequences degraded performance from overfitting.
> > >
> > > The paper has been updated to address some points made here. We also clarify the per-room statement about instructions, see Figures, 1,3 for examples.

---

> > > ### Author Response · Authors · 2021-11-20
> > > **Added Additional Baseline**
> > >
> > > We wanted to let the reviewer know that we added another baseline that just predicts the goal instead of the task instructions as requested. This can be found in the appendix in the same section as the other additional baselines.

---

### Official Review · Reviewer_EYpA · 2021-11-04

**Correctness:** 3
**Technical Novelty And Significance:** 2
**Empirical Novelty And Significance:** 3
**Recommendation:** 5
**Confidence:** 5

**Main Review:**

I believe this to be a well-written, strong paper, with an elegant and simple approach. The baselines go above and beyond, and though the evaluation is limited to BabyAI and Crafting simulated environments in 2D, I understand how these environments can be complex and multi-faceted, and serve as excellent testbeds.

I think the argument regarding the data regime is nuanced, but very very interesting, and is worth sharing with the community as well.
However, the one weakness I have is a pretty big one, and one that I hope the authors can discuss during the rebuttal; the availability of the intermediate language instructions (those used for auxiliary language prediction), and ensuring these intermediate language instructions are of the structure that’s necessary for the approach to work! It seems to be the case that these “auxiliary” instructions need to be pretty distinct from the goal, spelling out explicit subtasks, that cannot be directly inferred.

I’m curious where the authors think this data could come from when scaled up; when I look at the suggestions in the paper (exploiting PDDL planners in simulation environments — this seems debunked by the ALFRED experiments, or video captions — unclear that these have any instructional signal at all), I don’t really see the broader applicability of this approach. This feels like a strong, perhaps breaking assumption, which is really coloring my initial review.


Typos/Style/Questions:

Page 4: Langauge —> Language

**Summary Of The Paper:**

This paper explores a simple, elegant idea: the use of natural language as supervision when performing instruction following tasks. Centered around an evaluation of various tasks in the BabyAI complex grid world suite, as well as the Crafting environment introduced by Chen et. al (2021), the proposed approach takes an easy-to-digest tact: eat a sequence of observations from the environment with a Transformer, and predict two things: first, given a goal, predict the action to take (a traditional behavioral cloning objective). Second (and a core contribution of the approach) treat language prediction as an auxiliary task, predicting a sequence of language instructions that specify how to perform the task (which I liken to “subtasks” or some compact language describing an immediate outcome to optimize for, though the authors should please correct me if I’m misinterpreting!).

These language assumptions are assumed to come from an oracle (concretely, in the implementation, generated synthetically exploiting the simulated nature of the environments), though the authors argue that this sort of “guidance instructions” appears naturally in the wild; videos have captions, humans could provide this feedback cheaply, etc.

The bulk of the evaluation evaluates sample efficiency and performance on the long-horizon tasks in BabyAI/Crafting, comparing the proposed approach to several meaningful ablations (no auxiliary objectives, a hierarchical variant, a self-supervised contrastive variant, and the proposed approach). These baselines are incredibly well thought through, and serve as valuable comparison points for the proposed approach.

The final, perhaps salient point of this work is the data regime; the main body of the paper and the appendix both argue that the true value of this approach (language as auxiliary tasks) is when (1) we are operating in a moderate data regime — too few data, and we can’t learn decent behavior, too much data, and we can just learn tasks without additional supervision, and (2) when the supplemental language information is providing something new, over the goal information itself. Experiments on the ALFRED suite show that because intermediate language instructions can be predicted from the goal itself, they don’t have inherent value.

**Summary Of The Review:**

In general, I believe the proposed approach to be simple and elegant; the arguments about the data regime are well argued, and the evaluation is precise. However, my biggest concern has to do with the viability of obtaining the requisite auxiliary language instructions — I’m not convinced that this data can be easily obtained in the wild, which puts the value of this approach into question.

I sincerely hope that the authors can engage with me on this point!

---

> ### Author Response · Authors · 2021-11-11
> **Response to Reviewer EYpA**
>
> We thank the reviewer for characterizing our paper as “well-written” and “strong”. As the reviewer's main concern is with the viability of obtaining language instructions. We will discuss that here though we also encourage the reviewer to examine additional improvements to the draft.
>
> **On the viability of obtaining language labels:**
>
> 1. Data Collection Cost. When analyzing different forms of data, one must consider the cost of collection. Imagine that you are collecting demonstrations to teach a robot to complete the goal “get me a snack”. To collect the demonstrations, one would have to execute instructions like “go to the door, open it, … navigate to the refrigerator…” etc. At test time, we do not want to have to tell the agent exactly how to complete the task, but at train time we have already completed the instructions! Arguably, it seems like adding these instructions would not be the greatest challenge of data collection, particularly if you can do it simultaneously with data collection. Empirically, we show that language instructions can lead to a higher boost in performance than more demonstrations. For example, 25K Demos with instructions outperforms using 50K demos in BabyAI. When trying to design a high performance system the question should then be: What is easier, adding language instructions to a demo I was already going to collect, or collecting an entire additional demonstration? The broader applicability of our paper is to show the research community an alternative method of data collection which can be much cheaper depending on the situation. As we move beyond procedural generation and games, where demonstrations and instructions are both free with enough programming effort, we see this becoming a more important tradeoff. We have updated section 3.1 with some of these arguments.
>
> 2. The Amount of Data: Our method is focused on the low to medium data regime. Though there certainly could be benefits from collecting an image-net sized dataset of language instructions in some scenarios, that isn’t what we primarily investigate. We find that in low-data regimes specifically, where “scaling” is not the objective, language instructions can be better than demonstrations on the margin. Imitation learning itself is hard to scale due to the cost of demonstrations. Even if one were to believe that collecting demonstrations is easier than collecting language annotations, in table 4 we show that you can use far fewer language instructions than demonstrations and still get more performance. Vast scales of data would likely come from unsupervised or self-supervised approaches applied before fine tuning with imitation or language prediction.
>
> **Other Smaller points**:
>
> 1. *Using labels from PDDL:* First, as explained in our updated appendix, we find that the tasks in ALFRED are actually relatively simplistic. Empirically the instructions for completing a task (or the PDDL steps) can be recovered nearly perfectly from just the goal statement. If tasks become logically much more complex or involve more intermediate steps, like those in BabyAI, we would perhaps expect instructions derived from PDDL to reveal more unknown information (this is also discussed in bullet 1 in the above). Second, the BabyAI groundtruth planner considers finer resolution state changes, perhaps giving more signal. Regardless, we have updated the draft to make more careful suggestions about ways useful language instructions could be obtained from simulators in section 3.1.
>
> 2. *On Video Captions:*  We agree that videos sourced from the internet would likely require significant cleaning efforts before usable instruction data is obtained. We have clarified to draft to instead provide the suggestion that humans can collect data or play games while explaining what they are doing in section 3.1
>
> 3. *On Scaling to Harder Environments:* Our experiments demonstrate that a key ingredient for language prediction is difficulty. If tasks are too simple, language prediction is unlikely to be useful for high-level representation learning. We use similar environments to those used in other contemporary works on leveraging language instructions for hierarchical policies. Furthermore, many benchmarks considered to be extremely difficult in the status quo, like ALFRED, actually have logically simple tasks and are usually limited by visual understanding (See Appendix A.2) or use instructions themselves as inputs which can be cumbersome at test-time. We believe our paper serves as a call to the community to develop more combinatorially difficult and longer horizon benchmarks.
>
> In the light of the above clarifications, we would like to ask if you are willing to increase your score assuming we have addressed your concerns. Otherwise, please let us know if you have additional questions.

---

### Author Response · Authors · 2021-11-11
**Overall Response To Reviewers**

Dear Reviewers,

Thank you all for your reviews. We have responded to each review individually in more depth, but would like to make some overarching comments here. We are excited to engage with the reviewers!

First, we have updated the paper in the following ways:

1. Made changes to the introduction to better highlight the differences between instruction following and our setting. Additionally, we place more emphasis on ‘instruction’ prediction versus ‘language’ prediction.

2. Added further discussion of when language instructions are useful in appendix A.2 and completely overhauled the included experiments on ALFRED.

3. Added two additional baselines in appendix A.3

Parts of the paper that have changed can be found in blue, save for grammatical or single word edits.

Second, we would like to address the practicality of obtaining instructions, which a few reviewers have expressed concerns about. Data collection in any form is difficult but we believe instruction collection can actually be a relatively low burden.

1. If one has to collect real world demonstrations, the burden of collecting additional instruction labels at the same time seems low. We have updated section 3.1 to explain this. Furthermore, we show that adding language labels can be more effective than adding more demonstrations. This means the question should be: What is harder, collecting an instruction label, or collecting an entire additional demonstration? In many scenarios, we would argue that collecting an instruction label is easier. 1) if the dataset already exists, you may not be able to collect more demos, but could add instruction labels. 2) If demos require real-world diversity or have very long horizons, setting up more environments is likely very costly, making simultaneous instruction annotation an appealing option. 3) Even if you don’t face the costs of collection demos in the real world, if you can program an optimal policy in a simulator, the effort to have it additionally output instructions is likely marginal.

2. Our paper is focused on the low-data regime. We are not proposing the collection of vast amounts of language annotations, or scaling this approach to image-net sizes. In fact, we even show in table 4 that one doesn’t need to label all available demonstrations with instructions to see a boost in performance. In fact, having 50K demos and 25K instruction annotations performs within 3% of having 50K demos and all 50K instruction annotations in BabyAI.

Our work shows how useful instruction prediction can be, and we foresee behavior datasets in the future including more instruction or language labels.

---

### Decision · Program_Chairs · 2022-01-20

**Decision:**

Reject

**Comment:**

The reviewers all consider the paper to be below the acceptance bar. While the revision addressed some concerns, several critical ones remain open. This includes empirical concerns with regard to the extremely simple grid-world environments used, and with regard to the vague distinction between instructions and goal specifications. To improve the submission, the authors should seek stronger empirical foundations, and either refine or remove vague distinctions with regard to the phenomena they aim to study.

Special thanks to the reviewers for an extremely productive discussion.